# An Innovative Tool to Control Occupational Radon Exposure

**DOI:** 10.3390/ijerph191811280

**Published:** 2022-09-08

**Authors:** Lucía Martin-Gisbert, Alberto Ruano-Ravina, Juan Miguel Barros-Dios, Leonor Varela-Lema, Mónica Pérez-Ríos

**Affiliations:** 1Department of Preventive Medicine and Public Health, University of Santiago de Compostela, 15705 Santiago de Compostela, Spain; 2Cross-Disciplinary Research in Environmental Technologies (CRETUS), University of Santiago de Compostela, 15705 Santiago de Compostela, Spain; 3Health Research Institute of Santiago de Compostela (Instituto de Investigación Sanitaria de Santiago de Compostela—IDIS), 15706 Santiago de Compostela, Spain; 4Consortium for Biomedical Research in Epidemiology and Public Health (CIBER en Epidemiología y Salud Pública/CIBERESP), 28029 Madrid, Spain

**Keywords:** workplace, lung cancer, Directive 2013/59/Euratom, effective dose, ionizing radiation

## Abstract

After smoking, indoor radon is the main contributor to lung cancer in many countries. The European Union (EU) Directive 2013/59/Euratom establishes a maximum reference level of 300 Bq/m^3^ of radon concentration in the workplace, and an effective dose limit of 20 mSv per year for workers. If the radon concentration in a workplace exceeds the reference level, constructive mitigation applies. When constructive mitigation is not feasible, we propose to keep workers’ effective dose below 6 mSv per year (category B of exposed workers) by controlling occupancy time. Setting the maximum annual dose at 6 mSv protects workers’ health and eases the regulatory requirements for employers. If multisite workers are present, each worker has to be monitored individually by tracking the time spent and the radon concentration at each worksite. This paper shows a software tool for employers to perform this complex tracking in an accurate, conservative, and transparent manner, and in compliance with the EU by-laws.

## 1. Introduction

### 1.1. Radon and Lung Cancer

In many countries, indoor radon is the main contributor to lung cancer after tobacco smoking. The Earth crust naturally exhales radon as a result of the uranium radioactive decay. While radon gas remains at low concentrations outdoors, it can accumulate indoors, increasing the risk of lung cancer as the concentration and/or years of radon exposure increase [1]. There is a linear relationship between radon exposure and lung cancer risk, as shown by Darby et al. [2] and this association has also been observed for never smokers [3].

In order to enforce protection from radon exposure, Directive 2013/59/Euratom requires European Union (EU) member states to implement a maximum reference level of 300 Becquerels per cubic meter (Bq/m^3^) of radon concentration in homes and workplaces. If the radon concentration in the workplace exceeds the reference level, workers’ exposure shall be indicated as effective dose in millisieverts (mSv) [4]. The effective dose is proportional to the radon concentration and occupancy time in the workplace and it can be calculated using dose coefficients published by the International Commission of Radiological Protection [5].

In Europe, there are extensive radon-prone areas identified, namely based on residential measurements [6]. Available European surveys on occupational radon also showed that a relevant percentage of workers are exposed to concentrations of radon above the EU maximum reference level [7,8,9]. For instance, in a recent pilot study in Spain [10] including 245 occupational radon measurements, 27% exceeded the EU reference level (300 Bq/m^3^). Some of these excessive radon levels were also found outside radon-prone areas [10].

Nevertheless, several occupational radon surveys confirmed that radon-prone areas identified through residential measurements also do correlate with higher levels of radon in the workplace [7,9,10,11]. The EU directive also identifies specific workplaces where higher levels of radon are expected, such as mines, touristic caves, spas or water treatment facilities [12]. Some occupational radon surveys also observed higher levels of radon in old buildings used for cultural or recreational purposes [9,13].

It is noteworthy that workplace radon exposure can occur in combination with other exposures to ionizing radiation that may occur at home but also due to medical image testing. In general, the additive exposure of ionizing radiation should be considered, since workers exposed to high radon concentrations at work could be at an even higher risk of lung cancer if they have a high radon concentration at home combined with medical image testing. This means that exposure to ionizing radiation is being globally increased due to the use of artificial ionizing radiation [14].

### 1.2. EU Regulation: Generic Framework for Radon Protection in the Workplace

In order to ensure workers’ protection from radon and meet Directive 2013/59/Euratom requirements, employers can follow the EU guide “Radiation Protection No. 193” [15] and the pertaining National Regulator guides on radon protection, if available.

In summary, this is the generic framework for radon protection based on the EU guide [15] and complemented with some specifications from the Spanish Regulator guide on radon protection [16] (see Figure 1):Checking radon potential. There is radon potential if the workplace is occupied over 50 h a year and if it is located on one or more of the following locations [15]:Radon-prone area according to national radon maps;Underground, including basements, mines and caves;Thermal facilities and facilities for underground water treatment or storage;Buildings where high radon levels have been previously detected;Other workplaces specified in National Radon Plans or other national regulations.Measuring radon concentration. If there is radon potential, radon concentration should be measured following the competent authority guidelines [15]:If the concentration is above the national reference level, step 3 applies;If the concentration is below national reference level, measurement shall be repeated periodically as required by national regulations, and if there is any indication of a radon increase [15].Mitigation should be done following approved construction guidelines [16]. There are official guidelines published at the country level on how to perform adequate constructive interventions to mitigate radon, for instance, in the Technical Building Code (CTE) website in Spain [17], or in the *UKradon* website [18]. After mitigation, radon concentration should be measured to verify effectiveness [15]. If constructive mitigation indeed reduces the concentration below the reference level, step 2.b applies; otherwise, step 4 applies.If constructive mitigation cannot decrease radon concentration below the reference level after reasonable attempts, an operational mitigation can be done [15]. Before engaging in operational mitigation, the regulator may require adequate justification on why constructive mitigation is not feasible [16].Operational mitigation consists of managing workers’ exposure through dose surveillance [15]. Directive 2013/59/Euratom establishes two workers categories depending on their annual effective dose:Category B workers: Those exposed to 6 mSv or less per year must undergo “dose revision” to confirm they indeed do not exceed 6 mSv per year [4].Category A workers: Those exposed above 6 mSv require a “planned exposure” management that involves notification to the competent authority and the use of dosimeters in the workplace among other requirements. The effective dose limit is set at 20 mSv/year for all workers [4].Note: Pregnant and breastfeeding workers and those under 18 years old have specific dose limits (<1 mSv) [4].

### 1.3. Objective

The current situation underlines the need for actions to be taken to protect workers exposed to radon and comply with the EU directive. Constructive interventions to mitigate radon exposure are well described in several public guidelines at the national level. However, mitigation may be especially challenging for the employer when constructive mitigation is not effective or feasible, and an operational mitigation applies.

There are not clear guidelines on operational mitigation as there are for constructive mitigation. Furthermore, operational mitigation can become very complex if the employer manages multisite workers. In order to comply with the EU directive and protect workers, we aim to provide a tool for the operational mitigation of radon exposure in complex situations with multisite workers.

## 2. Methods

To develop the tool, we used a fit for purpose methodology which is usually employed for processes aimed at production quality improvement (from industrial to health procedures). The method is based on the plan-do-check-act quality improvement standards [19], and consists of defining the objective, generating the solution, checking it through experts and stakeholders’ reviews, and implementing it:“Plan” phase: The objective was defined through 4 face-to-face sessions between authors. The relevant bibliography and stakeholders were also identified in these sessions.The main deliverables of the plan phase were the generic framework for radon prevention in the workplace (see Section 1.2), the objective description (see Section 1.3) and the list of relevant bibliography (see references).“Do” phase: The solution proposal was generated based on: exhaustive review of selected regulatory and scientific references, active search and review of good practices within institutional websites from different countries, and ongoing face-to-face brainstorming sessions among authors.As a result, the main deliverable was a detailed document with a tentative solution proposal reviewed and approved by all authors. The list of stakeholders was also confirmed in this phase. The tentative solution included Excel sheets where different exposure scenarios were tested and reported.“Check” phase: The tentative solution proposal was sent for review to 4 stakeholders comprising the main types of stakeholders (workers, regulators, and employers). The solution proposal was checked sequentially as follows (Figure 2).Each stakeholder first received an informative phone call inviting them to participate. Once accepted, the document with the solution proposal was sent in Word format through email. Stakeholders were requested to review the document, add comments or correctios, and provide feedback on whether the solution was accurate, pertaining (would they use it or recommend using it?) and socially fair. Once the stakeholder sent their feedback, it was either directly included or discussed and then included or excluded upon agreement with the stakeholder. Then, the document was updated and sent to the next stakeholder.All stakeholders confirmed that the solution was accurate, pertaining and socially fair. Most of their comments and corrections where minor and did not affect the content of the proposal. The regulator feedback was the most relevant in content and was fully included in the solution proposal.The main outcome of this phase was the final solution proposal.“Act” phase: The solution proposal was materialized in a software application (see the Section 3) which was again tested and reviewed with the different stakeholders.

## 3. Results

When constructive mitigation is ineffective or unfeasible, we propose an operational mitigation that ensures the annual effective dose does not exceed 6 mSv by controlling the occupancy time. Keeping the effective dose under 6 mSv per year benefits both workers whose exposure remains relatively low, and employers because they reduce regulatory requirements and risks.

There are also complex situations where a single employer manages numerous multisite workers, meaning each worker spends a variable amount of time in different worksites with different radon concentrations as work demand fluctuates. In this case, each worker must be monitored individually and each exposure time at each workplace needs to be tracked. This tracking shall be accurate, conservative and transparent.

We developed a software tool for employers to perform this tracking. The software tool keeps track of each individual dose by tracking time and location in previously measured workplaces. This dose control facilitates ensuring that both the workforce planning and registration of the accumulated effective dose (mSv) are compliant with Directive 2013/59/Euratom.

The standard user of this tool is a health and safety technician or any professional in charge of handling worker safety for the employer, or the employer themself. The tool basic workflow consists of:Initial registration: This step only needs to be performed once unless there are new workers or work sites.oRegistration of the different work sites by introducing a name or ID for each site and its mean annual concentration of radon. oRegistration of the different workers by introducing an identifier and a name for each of them. Each worker’s accumulated dose is automatically updated by the tool and can be viewed at any time in this window. A report with all workers’ individual accumulated dose can also be downloaded at any time here.Compliant planning in four steps: These need to be followed for each job planned.Selection of work zone and work time as needed;Selection of workers from the list of “available” workers. Available workers are those that can perform the planned work without surpassing the established dose limits (set at 6 mSv per year). The worker dose is automatically updated during the planning. The tool would not allow a job being assigned to an “available” worker that would exceed 6 mSv after the job (e.g., if a worker has accumulated 5.6 mSv and the planned job would add 0.5 mSv, the tool would show an error box indicating that the selected worker cannot perform the job);Verification of the selected data: where the user can make one last check before completing the planning: once planned, the job data would flow to a dashboard;Confirmation of plan completion: Once the job has been completed, the user can edit the job in the dashboard if there was any change in the job plan (e.g., if the job took longer than expected, if an additional worker joined, if the worksite change). Any change in the plan would update workers’ dose count as applicable. If the job was carried out as planned, the user can just confirm that this was the case in the dashboard.Compliant reporting: Workers’ accumulated doses and the history of all planned jobs in the dashboard can be viewed at any time, and it can be downloaded for reporting purposes (for instance, quarterly or yearly for internal reporting) or in the case of inspection or upon request from a competent authority.

The software application (RadonPro) is being registered as intellectual property of the University of Santiago de Compostela and we are currently running demo sessions with potential users.

## 4. Discussion

We bring a tool for employers to handle complex scenarios of radon exposures with “multisite” workers. This tool allows smooth operational planning to maintain workers’ annual dose below 6 mSv and ensures regulatory compliance as dose history is automatically registered and easily retrieved in the case of regulator’s inspections. To date, we could not identify any similar solutions available.

The tool can be used as far as the employer can track and plan work time accurately, and if the mean annual radon concentration is known for each work area. This means that employers have followed compliant procedures to measure radon concentration in the workplace. The EU directive [4] established a period of at least two months of measuring radon in order to obtain the mean annual radon concentration of a worksite (or a home). There are also national guidelines that include specific procedures to measure radon concentration in the workplace [20].

The proposed tool has several strengths. This solution has been reviewed by relevant stakeholders from all relevant parties: worker representatives, employers and regulators, thus ensuring that this solution proposal is both fair and technically accurate. The tool eases regulatory compliance for employers and protects workers from radon exposure. Furthermore, it can promote transparency between employers (those responsible for workers’ safety), workers and competent authorities as individual dose records can be easily retrieved and reviewed. The tool also enables the optimization of human resources and improves workers’ safety, especially in workplaces with large differences in radon levels between sites. RadonPro ensures that effective dose is as evenly distributed as possible among workers by using time and site control, thus avoiding high accumulations of dose in a few individual workers.

This tool also has some limitations and challenges. The tool follows the latest EU regulatory framework [15] guidelines and dose coefficients from ICRP publication 137 [5], and the formulas were designed to be as conservative as possible. Thus, the occupancy time may be more restricted than requiredin some scenarios as workers’ safety has been prioritized. In the future, we contemplate an ad hoc validation process using individual dosimeters as controls, to provide data on the precision in which RadonPro performs effective dose calculation in a particular workplace.

In addition, both Directive 2013/59/Euratom and ICRP publication 137 may become obsolete in the long-term. This means that continuous review of state-of-the-art advances in occupational radon exposure is needed to keep the tool updated. This continuous review is possible as the tool is maintained as part of an internationally recognized research line on radon within the University of Santiago de Compostela (School of Medicine).

Unfortunately, this tool does not manage lung cancer risk arising from other sources: occupational exposures that may coexist with radon exposure, such as silica dust [21], or from non-occupational risks factors, namely smoking and residential radon. Thus, employers must identity and mitigate all occupational carcinogens, and on top of that, promote antismoking policies and support residential radon measurement and mitigation where applicable.

Although RadonPro is meant for occupational radon exposure, it could also include radon exposure at home as an input. This could be used by regulators or workers’ representatives to acknowledge the full individual risk arising from radon exposure. It could be especially relevant for workers living in radon-prone areas.

## 5. Impact on Global Goals

Finally, the provided solution brings a simple and compliant resource to reduce preventable lung cancer risk among workers, in line with Europe’s Beating Cancer Plan [22], the European Cancer Code and the United Nations Sustainable Development Goals 3 and 8.

## 6. Conclusions

Employers and authorities can, and must protect workers against radon exposure. The solution we propose here proofs it can be accomplished, even in complex exposure scenarios where reduction of radon concentration is not feasible, and multisite workers are present.

## Figures and Tables

**Figure 1 ijerph-19-11280-f001:**
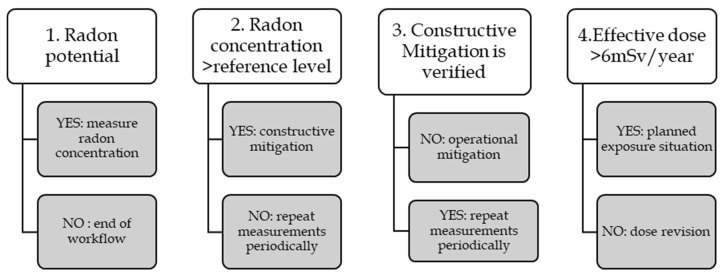
Workflow on how to ensure radon protection in the workplace.

**Figure 2 ijerph-19-11280-f002:**
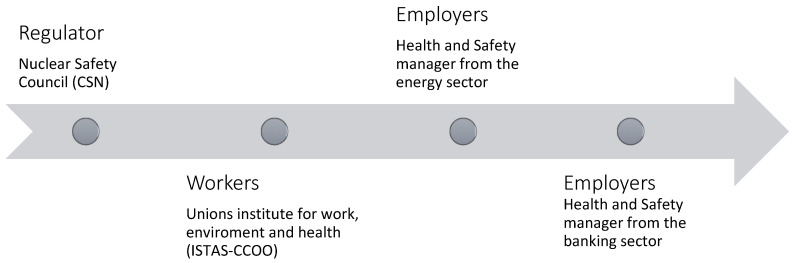
Sequence of stakeholders participating in the “check” phase.

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
