# Peer review of "An Innovative Tool to Control Occupational Radon Exposure"

_ijerph, 2022, doi:10.3390/ijerph191811280_

Round 1
Reviewer 1 Report
The manuscript does not contain any scientific issue, and it just intends to advertise an application which the authors are now developing. No data and application result are presented to confirm the authors' opinion. The manuscript is not acceptable as a scientific paper.
Reviewer 2 Report
I guess that this paper is a paper on the development of a software tool to perform ”RADIATION PROTECTION No.193” in Reference 5. However, it is not written that way in the abstract.
Most of what is written in the method seem to be shown in RADIATION PROTECTION No.193 of Ref. 5, but it is not specified in this paper. Figure 1 published on page 18 of Reference 5 is easier to understand than Figure 1 in this paper. It is necessary to specify Yes No and verification measurements in the figure. If category A worker and B worker are legally separated, it is necessary to specify that fact. It is necessary to rewrite the part quoted from other documents and the original part of the authors so that they can be understood.
